# Impact of the COVID-19 Pandemic on Psychosocial Well-being and Adaptation in Children and Adolescents with Asperger’s Syndrome

**DOI:** 10.3390/ijerph20032132

**Published:** 2023-01-24

**Authors:** Marta Tremolada, Mariangela Rosa, Roberta Maria Incardona, Livia Taverna, Michele Guidi

**Affiliations:** 1Department of Developmental and Social Psychology, University of Padua, 35131 Padua, Italy; 2Pediatric Hematology, Oncology and Stem Cell Transplant Center, Department of Woman’s and Child’s Health, University of Padua, 35128 Padua, Italy; 3Cooperativa Progetto Insieme, Via Cappello 44, Noventa Padovana, 35027 Padua, Italy; 4Faculty of Education, Free University of Bolzano-Bozen, 39100 Bolzano, Italy

**Keywords:** Asperger’s syndrome, atypically development, COVID-19, children, adolescents, typically development, parents, adaptation, well-being, multi-informant

## Abstract

Mental health could worsen in children and adolescents with autism spectrum disorder during the COVID-19 pandemic. In addition, their parents could be more at risk for an increase in anxiety and depression symptomatology. This study aims to understand the adaptation and the psychosocial well-being in a sample of 16 males aged 10–21 years old with Asperger’s syndrome after the quarantine for COVID-19 when they return to school and partially to their activities in September 2020. The adopted approach is multi-informant with a battery of questionnaires on psychological health and adaptation given by a secure online web data Qualtrics both to adolescents and also to their parents. Paralleling matched peers with typical developments were assessed by adopting the same methodology. The results evidenced several difficulties in psychological health in population with Asperger’s syndrome, especially in anxiety and socialization. Adaptation is put in crisis, even if they reported a good comprehension and adoption of the right behaviors anti-COVID-19. Parents of children with Asperger’s syndrome reported similar psychological difficulties and general health to those of the group of children typically developing. Some clinical indications could be discussed for psychologists who follow children and adolescents with Asperger’s syndrome.

## 1. Introduction

Autism spectrum disorder (ASD) is a neurodevelopmental disorder characterized by deficits in social communication and social interaction, in addition to restricted and repetitive patterns of behavior and interests. In 2013, the Diagnostic and Statistical Manual of Mental Disorders, 5th (DSM-5) [1] created the umbrella diagnosis of ASD, consolidating four previously separate disorders: autistic disorder, Asperger syndrome (AS), childhood disintegrative disorder, and pervasive developmental disorder not otherwise specified. Unlike previous versions, in DSM-5 more importance is placed on what is a developmentally oriented classification of childhood mental pathology, paying attention to the neurobiological etiology underlying these disorders. In this new classification, Asperger’s Syndrome (AS) is inserted into the Level 1 autism spectrum disorder, without intellectual impairment of the associated language [2]. The characteristics of the subjects with this syndrome are having special interests, reporting difficulties in social communication and social interaction, having an Intelligent Quotient > 70, and having early and formal language development but a lack of pragmatics of communication. Recently, data from the American Center for Disease Control (CDC) indicate at 8 years a prevalence of 1 out of 134 children [3], i.e., about 0.75%.

The World Health Organization (WHO) declared the new coronavirus (COVID-19) outbreak a global pandemic on 11 March 2020. It is a public health emergency of international concern and poses a challenge to psychological resilience [4]. It is necessary to understand the psychological impact that the epidemic itself and relative quarantine have on the entire population, especially children and young people with disabilities. The COVID-19 pandemic has led families to adapt their lives, including social isolation and working from home. The consequences of this outbreak of mental health are several. Change in routine is often a significant challenge for children with ASD [5], and for that reason, families with children with ASD could be a vulnerable group to develop anxiety and mental abnormalities during quarantine and isolation. Children with autism spectrum disorder (ASD) often experience changing routines as a major challenge. Results of a recent Italian study showed that restrictive measures had negative effects on children’s daily routines, which in turn was contributing to altered sleep habits, even if the parental self-efficacy perception remained stable during the home confinement compared to the period immediately before the lockdown [6].

For that reason, the need for adaptation during the COVID-19 pandemic may have brought great problems to families with children with this pathology. Children with ASD are at high risk for psychiatric problems during the COVID-19 pandemic, and the degree of understanding of the child of COVID-19, COVID-19 illness in the family, low family income, and depression and anxiety symptoms in parents increase the risk of poor mental health during the pandemic [7]. The results of recent studies showed a potentially important psychological impact of the COVID-19 pandemic not only in children with neurodevelopmental disorders but also in their caregivers, especially anxiety symptoms [8]. Parents of children with ASD had lower levels of resilience and more symptoms of anxiety and depression than parents of children typically developing [9].

The strict domestic quarantine policies adopted to control the transmission of COVID-19 could have adverse psychological effects and could exacerbate preexisting conditions such as depression and anxiety, especially in people with mental disorders [10]. Lockdown and boredom can reveal susceptibility to unhealthy behavior.

During quarantine, in addition to sociodemographic factors, the factors that seem to affect the worst psychological impact are: the duration of quarantine [11,12]; the fear of being infected and of being able to infect others [13,14]; boredom and frustration caused by the loss of daily routine and the reduction of physical and social contacts [12,15]; the lack of basic necessities (food, water, clothes) [16,17]; the scarce and inadequate information [15,18].

The numerous changes in the daily life of every single citizen with the lockdown have mainly been: school attendance, social and family relationships, also of children and young people both normal and with pervasive developmental disorders who have had to become aware of the health emergency.

In the adolescent population, some consequences in their health could be: sedentary behavior that increases proportionally to COVID-19 screen exposure [19], decreased physical activity time [19], increased insomnia and sleep-related problems [20] and increased difficulty falling asleep and sudden awakening episodes [21]. Additionally, from the point of view of psychic and psychological problems, adolescents seem to have been influenced by anxiety symptoms, and some studies show some influence on the anxiety levels of suicidal thoughts [22].

### Research Questions

Our interest focuses on the entire family unit of adolescents with Asperger’s syndrome who, like their healthy peers, have had to adapt not only to quarantine, especially to distance learning, but also to online rehabilitation activities during the period of isolation. There are limited studies on this population; for this reason, it is important to have a picture of their perceptions about risk and preventive or protective behaviors both retrospectively during quarantine and before the COVID-19 pandemic and in the current school reintegration and social life. Participants (children and parents) were asked to answer the questionnaires thinking about two different moments, that is the moment during quarantine (retrospectively) and after, during school reintegration.

The research questions are the following:

H1) Do children and young people with Asperger’s syndrome have a past and present good understanding of the risk of COVID-19 and of the preventive measures to contain it? We expect a good understanding and application of the anti-COVID rules in the function of the characteristic of this population that diligently respect the duties after that have been explained in a language they understand. For this reason, we presume that these children and adolescents respond correctly to both the knowledge and implementation of the anti-COVID rules.

H2) What were the possible psychological consequences, in children with Asperger’s syndrome and in their parents during quarantine?

In particular, we expect children and adolescents with Asperger’s syndrome to report greater well-being and a better perception of life during the lockdown, based on their social isolation and more focused attention on their interests.

We also expect that parents of children and young people with Asperger’s syndrome will show an increase in stress and the negative effects in terms of mental health (anxiety, depression and lower well-being) in the face of the accumulation of greater responsibilities on the academic-rehabilitative and job fronts.

H3) What have been the main difficulties for children and young people with Asperger’s syndrome in returning to school and resuming social life since September 2020? We expect, like the effects highlighted by the quarantine of boys with neurotypical development, in this clinical population the effects, extended over time, on an internalizing and externalizing level, specifically higher symptoms of anxiety, depression [23], and obsessive-compulsive disorder [24]. We expect lower well-being in children and adolescents with Asperger’s with the resumption of activities in September 2020 compared to the control group of healthy peers.

H4) Could the high parental stress experienced during the lockdown have influenced in the long term the emotional and behavioral difficulties their children have experienced with returning to school and everyday life since September 2020?

We expect, according to the effects and relationships between parental stress and emotional and behavioral symptoms of their children, investigated during the adaptation to the lockdown and the Coronavirus [21,25,26,27], a cumulative effect with their return to school, which we remember was not, especially for older children, a return to real everyday life, but rather an alternation between face-to-face school and distance learning.

## 2. Materials and Methods

### 2.1. Procedures

After obtaining the consent of the Ethics Committee in Psychology (Protocol 3814, University of Padua) on 1 November 2020, participants were recruited. It is necessary to distinguish the recruitment of families with children with Asperger’s syndrome from families with children with neurotypical development, aged between 9 and 20 years. The first group was reached through a letter of presentation of the project to the President of the Asperger Veneto Regional Association and by the collaboration of Dr. Guidi as head of the center of the province of Padua. The control group was recruited using a snowball procedure. The clinic and the control group signed the informed consent form before filling in the protection online questionnaires on the Qualtrics platform.

### 2.2. Participants

Sixteen school-age children aged 10 to 20 years with Asperger’s syndrome attended school and were recruited (M = 14; SD = 3.62); while young adults were not included in this study. At the same time, neurotypically developing peers (M = 14.07, SD = 3.31) of the same sex, region and age as close as possible were also recruited.

Table 1 shows the characteristics of the participants.

### 2.3. Instruments

The methodological approach of this study is mixed (explanatory sequential design) in which the quantitative analysis is deepened by the qualitative one; and multi-informant (involvement of parents and their children and young people).

The instruments for children/adolescents are several. They were filled in with the supervision of a psychologist that clearly explained the instructions during two meetings with a global duration of about 40 min. They are the following: Percieved Risk Questionnaire, Cantrill’s Self-Anchoring Laddaer of Staisfaction, Safa battery.

#### 2.3.1. Perceived Risk Questionnaire

It is an ad hoc questionnaire, divided into two age groups (10–14 years and 15–20 years), with 42 statements on contagion, risks and knowledge of behaviors both adopted in the past, during the quarantine, and promoted today in the post-isolation, to which they were asked to answer with true or false. Statements were divided into four subscales: information, mode of transmission, symptoms and behavior.

#### 2.3.2. Cantrill’s Self-Anchoring Ladder of Life Satisfaction

It is a tool for measuring people’s attitudes toward their life and its components in various respects [28]. Participants are presented with three visual scales numbered 0–10 and told the following statement: “Could you please indicate, on a scale from 1 to 10, where 1 represents the worst possible life and 10 the best”:What level would you have been at about a year ago when you started school in September 2019.At what level would you have placed yourself during quarantine.At what level do you position yourself at the moment.

Of the following scales, three levels of life satisfaction were defined: low (0–6), medium (7–8), and high (9–10). In particular, as regards the first two statements, the students are asked to remember a moment of life from the more remote (first statement) and more recent (second statement) past.

#### 2.3.3. The SAFA Battery

The SAFA battery is a self-administration diagnostic tool, it is adapted to three age groups: 8–10 years, 11–13 years and 14–19 years [29]. This favors the reliability of the instrument in relation to age.

The instruments scales, which present a textbook internal consistency index (α of Cronbach) in general higher than 0.80 (very good), can also be administered individually, this allowed us to use only the subscales for the study: “Anxiety”, “Depression” and “Obsessions”; for a total of 110 items for the younger age group and 144 items for the remaining two age groups. All participants were asked to respond with a true, false, or somewhere-in-between statement.

The instruments for parents are the following: ABAS-II, CES-D, GAD7 and GH12.

#### 2.3.4. The Adaptive Behavior Assessment System II (ABAS-II)

It is a scale for assessing children’s daily living skills compiled by parents [30]. For our study, having a group of participants who are not yet workers, we did not take into consideration the “Work” subscale; while of the remaining nine subscales, there were three conceptual, practical, and social domains. We administered the entire subscales of the practical domain (“Self-care”, “Home/school life”, “Use of the environment”, “Health and safety”) and the social domain (“Leisure”, “Socialisation”) and, finally, only one of the three subscales of the conceptual domain (“Self-control”) for a total of 164 items.

Parents were asked to respond on a 4-point scale, ‘0-Not able, 1-Never able, 2-Sometimes able, 3-Always able”; finally, to avoid causing confusion in the participants, we excluded the possibility of choosing the answer ‘I suppose’, even in the face of the physical absence of the administrator who could not have helped the parents with the compilation.

#### 2.3.5. Center for Epidemiological Studies-Depression (CES-D)

It is a self-assessment scale on depression, measured by 20 items (of which four reverse items) on a 4-point scale (0-”Never very rarely (less than a day)”, 1-”Occasionally (1 or 2 days)”, 2-“ Very often (3 or 4 days)”, 3-“ Frequently, all the time (5 to 7 days)”); with clinical cut-off equal to 16 [31]. Specifically, a total score between 0 and 9 indicates the absence of depressive symptoms and mild symptoms with a score between 10 and 15; however, it becomes relevant to identify moderate depressive symptoms (score between 16 and 24) or severe symptoms when the total score is greater than or equal to 25. Cronbach’s alpha of 85 is excellent evidence of the high reliability of the tool’s content.

#### 2.3.6. General Anxiety Disorder-7 (GAD-7)

It is a self-administered questionnaire used as a screening tool and as an assessment of the interference of symptoms of generalized anxiety disorder, in retrospective mode, consisting of 7 items [32]. There are 4 levels of presence/frequency of symptoms during the last two weeks (a score is then assigned to each level). Never present (0 pts), Present a few days (1 pt), Present more than half of the days (2 pts), Present almost every day (3 pts). The final scores of 5, 10, and 15 correspond to the cut-off points to define mild anxiety disorder, moderate anxiety disorder, and severe anxiety disorder, respectively. Finally, Cronbach’s alpha of.80 shows a good test of the reliability of the instrument’s content.

#### 2.3.7. General Health Questionnaire (GH12)

Participants were asked to complete the General Health Questionnaire (GH12), a validated questionnaire of 12 items assessing quality of life. The GH12 [33,34] provides a reliable, valid and brief assessment of quality of life. The 12-item self-rated questionnaire measures physical health, psychological health, social relationships, and the environment during the past two weeks. Each item is rated on a 4-point Likert scale (“less than usual”, “no more than usual”, “rather more than usual”, “much more than usual”), referring to the past 2 weeks, with a range of total scores from 0 to 36. Higher scores indicate a more problematic situation and there are different cut-offs to consider: Scores 15 are viewed as “problematic” and scores 19 as “very problematic”. An example item is: ‘Able to concentrate’. Overall, the higher total scores on the GHQ indicate a higher psychological discomfort experienced by the respondents and vice versa. GH12 has good to excellent reliability psychometric properties and performed well in preliminary validity tests that were also carried out in the Italian population [35,36]. Furthermore, the tool was recently used with adults during lockdown [37], proving to be valid and reliable (α = 0.80).

### 2.4. Statistical Analysis Plan

To answer the research hypotheses, statistical analyzes were performed using IBM SPSS Statistics 27 statistical software, (IBM corp, Armonk, NY, USA). Initially, descriptive statistical analyses of the sociodemographic variables of the entire group of participating children and their parents were carried out separately.

Precisely, given the small number of participants, it was decided to use nonparametric analyses with the aim of obtaining statistically more robust results. In particular, the Wilcoxon ranks test for paired and dependent samples, the Mann-Whitney test with two independent samples, the two-way Spearman correlation, repeated measures ANOVA and the Chi-square test were used.

## 3. Results

### 3.1. Knolewdge about Coronavirus and Anti-COVID Rules in Asperger and Typically Adolescents

Paired-sample Wilcoxon tests were run to see the possible differences between right and wrong answers in the two groups. The mean ranks within the groups were significant, both for the clinical group (Z = −3.50, *p* < 0.001) and for the control group (Z = −3.43, *p* = 0.001). The entire group of participants analyzed the answers, on average, correctly to a greater number of questions.

On the other hand, the comparison between the means of the ranks of the two paired samples of correct answers between groups appears to be insignificant (Z = −1.89, *p* = 0.85). Thus, we demonstrate how the boys in both groups respond correctly on average to the same number of questions on the COVID-19 test.

### 3.2. Possible Psychological Consequences in Children with Asperger’s and in Their Parents during Quarantine

First, a paired sample Wilcoxon test was performed to identify possible differences in perceptions of life between the two groups. The two groups did not show differences in their life perception scores (Z = 0.13, *p* = 0.90), showing an insufficient one in 53.3% of cases, a sufficient one in 26.7% and a very good one in 20% of cases.

In the clinical group, on a statistical level, no significant differences emerged between the scores attributed to one’s own life between T1 (life in September 2019, before COVID-19 pandemic) and T2 (Life during quarantine) (Z = −1.44, *p* = 0.15); between T1 and T3 (Life-related to the school returning on September 2020) (Z = −0.99, *p* = 0.32); and between T2 and T3 (Z = −1.21, *p* = 0.23). On the other side, in neurotypical children, no significant differences emerged between the scores attributed to one’s life between T1 and T3 (Z = −1.59, *p* = 0.113); instead, we find significant differences between T1 and T2 (Z = −2.12, *p* = 0.034) and between T2 and T3 (Z = −2.29, *p* = 0.022). These results, related to the neurotypical group, were investigated using repeated measures ANOVA, in which the data did not satisfy the sphericity hypothesis of the Mauchly sphericity test [W(2) = 0.61, *p* = 0. 04], thus using the Greenhouse-Geisser correction criteria, a significant difference emerged between the three Cantrill scales in the different periods investigated [F (1.44) = 5.47, *p* = 0.020], which can be observed in Figure 1.

Then, the Wilcoxon rank test was run on paired samples to understand whether parents with Asperger’s children experienced the same stress and worsening well-being as parents with neurotypical children. <this was followed because there were no significant differences in depression (Z = −0.347, *p* = 0.729), anxiety (Z = −0.918, *p* = 0.359) and well-being (Z = −1.06, *p* = 0.291) between the two groups of parents (clinical and control). See Figure 2a for the clinic group and Figure 2b for the control group.

### 3.3. Main Psychological Difficulties for Children and Young People with Asperger Syndrome with Returning to School and Resuming Social Life since September 2020

The Wilcoxon rank test for paired samples was calculated. The aim was to evaluate the presence of a significant difference in anxiety, depression, and obsession scores between the clinic and the control groups.

The comparison between the rank means of the two matched samples was significant only for the SAFA subscale related to anxiety (Z = −2.59, *p* = 0.01), where adolescents with Asperger syndrome (clinic group) reported, on average, higher scores (Mean = 8.12) compared to those neurotypical (control group) (Mean = 7.25). Figure 3 shows the placement at the normal or clinical level for both groups on all SAFA scales.

**Figure 2 ijerph-20-02132-f002:**
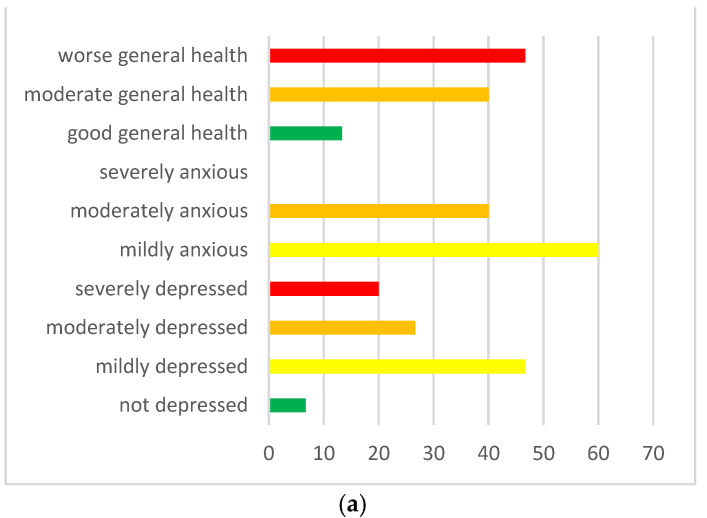
(**a**) Parents symptomatology in the clinic group. (**b**) Parental symptomatology in the control group. Legend: red = clinic score; orange = moderate clinic score; yellow: mildly clinic score; green = normal score.

Parents of children and adolescents with Asperger’s reported worse adaptive behaviors compared with healthy peers, especially the difficulties in socialization tasks, the identified worst adaptive behavior area (Figure 4).

Only secondarily, a series of two paired samples Wicoxon test were run to analyze whether there was a significant difference in each adaptive behavior scale along the belonging on clinic or control group. Differences were found for the General Adaptive Composite GAC (Z = −2.55; *p* = 0.01), Conceptual Composite CC (Z = −2.13; *p* = 0.3), Social Composite SC (Z = −3.11; *p* = 0.002) and Practical Composite PC (Z = −2.36; *p* = 0.02).

The clinic group members had significantly lower averages in GAC (M = 60.43; SD = 12.92) than the second group (M = 74.12; SD = 10.58), in CC (M = 55.22; SD = 5.33 versus M = 61.17; SD = 3.56), in SC (M = 65.13; SD = 11.03 versus M = 83.38; SD = 14.88) and in PC (M = 69.33; SD = 20.22 versus M = 87.25; SD = 12.67).

### 3.4. The High Parental Stress Experienced during the Lockdown Influenced the Emotional and Behavioral Difficulties when Children Experienced the Return to School and to Everyday Life since September 2020

For the Asperger group, a significant correlation was found between parental depression scores during lockdown and children’s anxiety subscale scores related to school return since September 2020 (r(13) = 0.56, *p* = 0.029). For the neurotypical group, a significant correlation was found between the questionnaire scores on the well-being perceived by their parents during quarantine and the scores related to the perception of life that adolescents themselves attribute to the “current moment” (r(13) = −0.57, *p* = 0.02).

## 4. Discussion

What motivated this study was to try to give a greater voice to children and young people with Asperger syndrome (clinical population still little investigated from the point of view of the literature) with regard to their perceptions and evaluations at a retrospective level during the lockdown and their understanding of the anti-COVID rules. Above all, we wanted to investigate the long-term consequences associated with the resumption of everyday life. Furthermore, taking into consideration the parental point of view and comparing it with those of neurotypical children, the secondary aim would be, through a comparative analysis, to prepare ad hoc guidelines for parents and teachers on how to support and help children with Asperger’s, the behaviors to follow and the management of safe and secure behaviors in schools and in relationships with others; but also to give feedback to the operators of their centers of afference.

The psychologists of the center had underlined a notable characteristic of Asperger’s children, namely the diligent respect of rules and duties after they had been explained in a language they could understand. These explanations were given not only by the psychologists and parents but also by the school, which assumed a fundamental role in the prompt provision of various information on COVID-19 and in the support of adolescents. This feature allowed hypothesizing an adequate knowledge and constant implementation of the anti-COVID rules confirmed by the results of the ad hoc questionnaire on perceived risk. In fact, the data confirm a good understanding and application of the regulations by the Asperger’s participants on the same level as their neurotypical peers, precisely out of a total of 42 questions both groups correctly answered on average 35 questions. In particular, the former correctly answered 83.6% of the questions, while the neurotypical peers 84.3%.

Regarding the second area of investigation, aimed at identifying the consequences, in terms of perceived stress, of family members of the population with Asperger’s syndrome during the quarantine; two research questions were analyzed separately. Our expectations regarding the greater well-being of Asperger’s children in quarantine were only partially confirmed. In fact, from their point of view, the perception of life in quarantine is estimated, like neurotypical peers, in 53.3% with an insufficient vote; this tends to fade in the post-quarantine period, however, remaining higher in Asperger’s boys (33.3%) than in neurotypical boys (26.7%). In fact, adolescents in the experimental group, on average, as those in the control group, perceived greater discomfort during quarantine, not benefiting as uniformly from social distancing as those with neurological disorders [38]. Although more in line with our expectation relating to the third research question, which foresees greater difficulties for the clinical group from September 2020, Asperger’s children perceived their life more negatively than their neurotypical peers, precisely in the period of the resumption of daily activities such as, for example, school in attendance.

Although the perception of life during quarantine has an insufficient assessment for just over half of Asperger’s children, it is not excessively high since, at the individual and family level, functional strategies have been implemented to better deal with the pandemic situation. On the one hand, adolescents uniformly declared that they perceived benefits in terms of limiting travel and extracurricular activities by obtaining more time for themselves by focusing on their special interests; on the other hand, parents have tried to reconstitute stable and positive routines such as moments of family conviviality during meals, “virtual workouts” and activities games, becoming important protective factors for the entire family unit [39,40,41]. Furthermore, these new routines have made it possible to better regulate and discharge the emotionality of the children, thanks to the active and moderate participation of the parents themselves [42] but also to reduce two anomalous behaviors of the children reported in the lockdown questionnaire, such as irritable behavior and lack of appetite.

Parents of atypical development [27] reported, as parents of children with neurotypical development [42], negative effects in terms of increased stress and worsening mental health. This hypothesis was fully corroborated. In fact, in both groups, it was possible to highlight symptoms, although not excessively elevated, of anxiety and depression (higher frequency between mild and moderate), and in most parents, a deterioration of well-being during quarantine. Furthermore, no significant differences were identified between the symptoms of the parents and the age groups of the children and also regarding the symptoms between the two groups of parents.

The investigation of the third research area on the difficulties in Asperger’s children identified with the resumption of everyday life from September 2020 only partially confirmed our hypothesis. In fact, as already expressed above, children with Asperger’s syndrome evaluate their lives by referring to the present moment with an insufficient vote in 33% of children, while the remaining percentage has medium-high evaluations. This lack of homogeneity of response can be explained by the fact that Asperger’s children, while expressing, in most of the interviews, happiness in returning to school and in resuming activities in the center of reference, however, compared to their neurotypical peers, they constantly experience a high sense of uncertainty linked, albeit minimally, to having to become used to face-to-face activities and, moreover, to the possibility of returning to DaD and therefore in quarantine or of being able to contract COVID in the face of a still deep-rooted health emergency.

These concerns are revealed by the presence, in Asperger’s children, of depressive (37.5%) and anxious (25%) symptoms related to the possibility of returning to quarantine and obsessive (25%) symptoms related to the possibility of contracting the virus, which leads them to implement, and also ask others to respect, all the anti-COVID rules (wash your hands, do not touch other people’s objects, then the mask or your eyes, etc.). Specifically, all three symptoms are higher than in the control group; especially anxiety does not appear to be clinical for the neurotypical sample; while in both groups, obsessive symptoms appear to be higher in boys between 15 and 20 years of age.

This symptomatology supports the results of studies conducted on neurotypical and neurological adolescents, in which direct effects of COVID related to anxiety, depression [23], and obsessions [24,38]; in the quarantine period and immediately after. It was hypothesized that these consequences could also be found in the long term, an aspect traced in our study with reference to the Asperger’s population and the period from September 2020.

The results obtained for the fourth and last research area confirmed partial cumulative effects of parental stress during quarantine and behavioral and emotional imbalance on return to school from September 2020, in the knowledge that parental stress had already negatively influenced their own children [21,25,26,27]. Specifically, it seems that in children with Asperger syndrome the greater depressive symptoms of the parent during quarantine, also due to the difficulties in reconciling work and support for the DaD, have a more negative influence on anxiety symptoms (albeit moderate) related to the return to daily life in children between the ages of 10 and 14. Probably, the guys from this age group, also in the face of a greater possibility of attending their own institutions in person more than older children, could perceive more anxiety related to a situation of returning to DaD and “locked up at home” by taking siblings as an example or rather older friends, compared to a greater “resignation” to the situation for the latter.

A narrative of an adolescent explains well this concept: “We really wanted to go back to school, I was delighted to see everyone again, with a little fear that we might attack COVID”—“At first I was intrigued about what it would be like to go back to school and I was dubious about what it would be like to do it with masks… it went well and I was happy!” —“I was delighted to go back to class; however, it took me some time to get used to the situation again. It has been a beautiful few months but we were all always anxious for the other risk of being able to return to DaD from one day to the next, as in reality then it happened!”.

In neurotypical adolescents, lower well-being perceived by parents during quarantine correlates with a lower evaluation that the boys themselves give to their lives in the current moment; in particular, it seems to be the older boys who express these evaluations, thus perceiving even greater effects of the well-being experienced in quarantine by their parents.

### Strengths and Limits

One of the limitations of this research is certainly the number of participants and the gender and age characteristics of the sample, which do not make the results sufficiently generalizable. The sample made up of 16 Asperger’s children paired with neurotypically developed children must, however, take into account both the niche clinical population that we have tried to recruit in the project and the historical context in which entire families have found themselves facing the overwhelming and disharmonious everyday life, which could have undermined the greater adherence to research. In this sense, a more comprehensive picture cannot be formed, unfortunately, due to the extremely limited number of staff. The results tend rather than show clear differences.

An aspect of particular relevance is the fact that the participants are all males and there is little heterogeneity as regards their age and distribution in the three academic ranges (primary, lower secondary, and upper secondary schools). It would be interesting to extend the survey also to girls with Asperger’s syndrome, albeit of lesser diagnostic importance precisely because they are very often underdiagnosed and at the same time make the age groups of the participants more homogeneous.

A second important limitation was the large number of items to which the boys were subjected; specifically, the application of three subscales of the SAFA (anxiety, depression and obsessions) made it more difficult to maintain the attention of the participants and lengthened the time required to complete the questionnaire, especially for the adolescents in the experimental group.

Finally, the predominantly transversal data collection that refers in many respects to the present moment, integrated with specific questions and questionnaires in which they are asked to recall aspects related to the more remote past (September 2019) and more recent (during quarantine) has encountered considerable difficulties in the Asperger population regarding the temporal aspect of the past.

Despite these limitations, we can identify three strengths of the research: the first is related to the fact that the research itself is the first, to our knowledge, to investigate difficulties during quarantine but, above all, the long-term effects of the measures determined from the onset of the Coronavirus pandemic in Asperger’s children and teenagers and the perceived difficulties with returning to everyday life. A second merit is that we conducted the survey on families in Veneto, one of the most affected areas on the national territory, to highlight the consequences in a context in which there has been a greater incidence of COVID. With reference to the latter aspect, it would be interesting to be able to estimate the presence of differences that we could find in families with the same atypical population but in central-south regions of Italy.

Finally, a last and great advantage is highlighted by the use of the mixed and multi-informant approach. The first allowed us to delve into the narrowest niche of Paduan families, aspects more generally investigated through questionnaires, both regarding Asperger’s children and their parents; the second, through the involvement of significant family members, has allowed us to better enrich not only the individual but also the family picture.

From a future perspective, not too distant, it could be interesting to include support teachers in the study, with the aim not only of investigating their experiences in quarantine with the DaD and the fluctuating and partial recovery of the presence for most of the new academic year; but also of highlighting their role for Asperger’s children and highlighting aspects of greater management difficulty that could be supported by ad hoc guidelines.

Furthermore, for future studies, it would be more necessary to reduce the workload required for the recruited clinical and neurotypical population.

Finally, starting from our results, it could be interesting to investigate in the same group of participants one year from today whether and for how long the symptoms persist, making this study a starting point for a possible longitudinal study of the consequences of a pandemic that has marked, like a scar, each of our experiences.

In future studies, it should be interesting to find and discuss also coping strategies and solutions that some families used to overcome the disease to have suggestions for clinical work with those children and their families.

## 5. Conclusions

This study focuses on psychosocial wellbeing and adaptation in children and adolescents with Asperger’s syndrome and their parents associated with the COVID-19 pandemic. It adoptes a multi-informant approach to have information on the specific needs of this clinical population. 

The main results evidence a good comprehension and adoption of the behaviors anti-COVID-19 in children and adolescents with Asperger’s syndrome, even if they report a clinical symptomatology in anxiety and in socialization and difficulties in several domains of adaptation. The guys with Asperger’s syndrome aged 10–14 years old, also in the face of a greater possibility of attending their own institutions in person more than older children, could perceive more anxiety related to a situation of returning to DaD and “locked up at home”. 

However, parents of the clinical group reported a similar trend in their psychological health comparing with those belonging to the control group, underlying how the negative psychological effects of COVID-19 are equally distribuited.

Clinical considerations could be taken basing on these empirical results.

## Figures and Tables

**Figure 1 ijerph-20-02132-f001:**
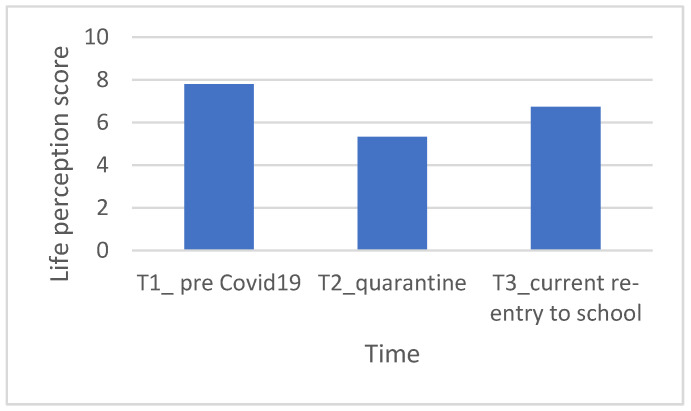
Estimated marginal means of Cantrill’s life perceptions in the neurotypical group. Legend: T1 = Life September 2019; T2 = Life in quarantine; T3 = Life-related to return to everyday life/September 2020.

**Figure 3 ijerph-20-02132-f003:**
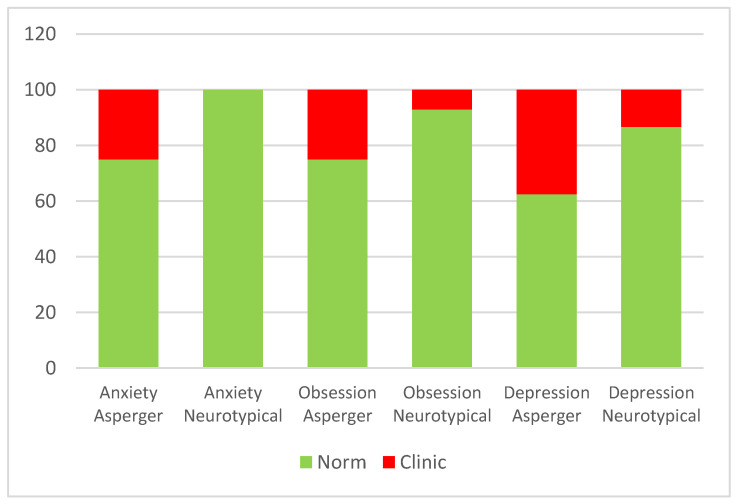
Placement at the normal or clinical level for both groups on all SAFA scales.

**Figure 4 ijerph-20-02132-f004:**
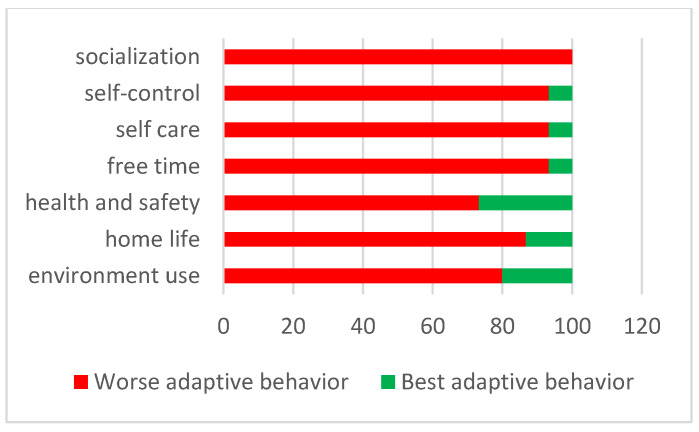
Placement in the worst or best adaptive behavior in ABAS-2 scores in the clinic group.

**Table 1 ijerph-20-02132-t001:** Sociodemographic characteristics of the participants.

Asperger					Neuro Typical		
Adolescents		Frequency	Mean	SD	Frequency	Mean	SD
Gender	Male	16			16		
Age			14	3.62		14.07	3.31
Age groups	10–14 yrs old	10			9		
	15–20 yrs old	6			6		
School	Primary	3			3		
	Secondary first grade	6			5		
	Secondary second grade	7			7		
Support school	Yes	12			0		
	No	4			15		
Parents		Frequency	Mean	SD	Frequency	Mean	SD
Role	Father	4			4		
	Mother	11			12		
Education	Primary/secondary school first grade	0			0		
	Secondary second grade	6			9		
	Yunior degree	2			1		
	Degree	7			5		
	Master/Ph.D.	3			1		
Number of children	1	4			4		
	2	8			9		
	3	2			2		
	4	0			1		
	5	1			0		
Child with other disabilities	Yes	2			0		
	No	13			15		

## Data Availability

The data presented in this study are available on request from the corresponding author.

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
