# Peer review of "Impact of the COVID-19 Pandemic on Psychosocial Well-being and Adaptation in Children and Adolescents with Asperger’s Syndrome"

_ijerph, 2023, doi:10.3390/ijerph20032132_

Round 1

Reviewer 1 Report

Summary

The present paper aims at investigating the consequences on the psychosocial well-being and adaptation of the quarantine for COVID-19 in small sample of children with Asperger’s syndrome compared with neurotypically developing children. Results show difficulties in Asperger’s psychological health and adaption. Further, parents of children with Asperger reported similar psychological difficulties and general health than those of the group of children typically developing.

From my point of view, the topic is interesting, and the theoretical background is sufficiently thorough.

However, there are some concerns that prevent me to recommend the publication of the paper in its current form.

Here is a list:

1)    Title: In the title the word “during” is misleading and should be changed. This because the present study takes into account the situation “after the quarantine for Covid-19, when children return to school and partially to their activities in September 16 2020 (as it is reported in the abstract, lines 15-16).

2)    Recently it has been published an interesting work that investigates the impact of the COVID-19 lockdown on the psychological well-being in a sample of Italian children with Fragile X syndrome and their families. It is well known that FXS children showed frequent autistic-like behaviors and it could be useful for authors to mention such paper (Di Giorgio, E., Polli, R., Lunghi, M., & Murgia, A. Impact of the COVID-19 Italian Lockdown on the Physiological and Psychological Well-Being of Children with Fragile X Syndrome and Their Families. Int. J. Environ. Res. Public Health 2021, 18, 5752. https://doi.org/10.3390/ ijerph18115752).

3)    Pag.3, lines 86-87 “and some studies show some influence on the anxiety levels of suicidal thoughts”. Please add a citation.

Research questions paragraph

4)    P. 2, lines 90-94 “There are limited studies on this population, for this reason, it is important to have a picture of their perceptions about risk and preventive or protective behaviors both during quarantine retrospectively and in school reintegration and social life”. This sentence suggests that the participants (children and parents) were asked to answer the questionnaires thinking about two different moments, that is the moment during quarantine (retrospectively) and after, during school reintegration. Is it correct? However, results are not in line with this methodology. Therefore, authors should specify what are the instructions for participants. Further, they should also specify the total time taken to complete the questionnaires.

5)    It could be useful that authors specify their expectations about each research question.

Instruments paragraph

6)    In the present version of the manuscript, it is not clear who fill out each questionnaire: children, parents, both? Here some examples: “They” for the Perceived risk questionnaire, “Students” (p. 3, line 142) for the Cantrill’s, “all participants” (both parents and children?) for the SAFA, and so on.

Results

7)    P.6, lines 226-234. I’m a little bit confused about what T1, T2 and T3 means. First, the questionnaire to which the authors are referring should be specified, and then also to this distinction between timepoints (see also the related point 4).

8)    P. 6 line 256, p.7 lines 257-258. “Neurotypical adolescents (control group?) reported, on average, higher scores compared to the control group (?)”. There is something wrong in this sentence. Neurotypical adolescents = control group vs. adolescent with Asperger’s syndrome = clinical group. So, please check this sentence.

9)    P. 8, lines 267-269 “However, in terms of the adaptive behavior of Asperger's children assessed by their parents, the worst adaptive behaviors were several, especially the area of socialization, 268 evaluated by all parents as the area with the worst adaptive behavior (Figure 4)”. English check is needed.

10) In general, graphics must be improved.

Discussion

11) P.8, line 296 “taking into consideration the entire family unit” I’m not sure that this is true, because usually who complete the questionnaire (online) is the mother (and children for this specific study), unless the instructions given to the participants were that both parents should complete the questionnaires (see point 4).

Author Response

Here is a list:

  • Title: In the title the word “during” is misleading and should be changed. This because the present study takes into account the situation “after the quarantine for Covid-19, when children return to school and partially to their activities in September 16 2020 (as it is reported in the abstract, lines 15-16).

The title has been changed following your suggestion

2)    Recently it has been published an interesting work that investigates the impact of the COVID-19 lockdown on the psychological well-being in a sample of Italian children with Fragile X syndrome and their families. It is well known that FXS children showed frequent autistic-like behaviors and it could be useful for authors to mention such paper (Di Giorgio, E., Polli, R., Lunghi, M., & Murgia, A. Impact of the COVID-19 Italian Lockdown on the Physiological and Psychological Well-Being of Children with Fragile X Syndrome and Their Families. Int. J. Environ. Res. Public Health 2021, 18, 5752. https://doi.org/10.3390/ ijerph18115752).

  1. We added this quotation in the main text (lines 58-62) and in the references (6). Thank you for your suggestion

3)    Pag.3, lines 86-87 “and some studies show some influence on the anxiety levels of suicidal thoughts”. Please add a citation.

We added it. See lines 93 and quotation 22

Research questions paragraph

4)    P. 2, lines 90-94 “There are limited studies on this population, for this reason, it is important to have a picture of their perceptions about risk and preventive or protective behaviors both during quarantine retrospectively and in school reintegration and social life”. This sentence suggests that the participants (children and parents) were asked to answer the questionnaires thinking about two different moments, that is the moment during quarantine (retrospectively) and after, during school reintegration. Is it correct? However, results are not in line with this methodology. Therefore, authors should specify what are the instructions for participants. Further, they should also specify the total time taken to complete the questionnaires.

We clarified the questionnaires’ instructions both in the research question section and in the procedure (lines 137-141). Results reflected the research questions that we had clarified (see lines 143-177) . We added the instructions and the total time to complete questionnaire in the procedure section (lines 200-202)

5)    It could be useful that authors specify their expectations about each research question.

We added them. See lines 143-177

Instruments paragraph

6)    In the present version of the manuscript, it is not clear who fill out each questionnaire: children, parents, both? Here some examples: “They” for the Perceived risk questionnaire, “Students” (p. 3, line 142) for the Cantrill’s, “all participants” (both parents and children?) for the SAFA, and so on.

We added a sentence that clarified the questionnaires given to children and adolescents and those given to parents. See line 200 and 236.

Results

7)    P.6, lines 226-234. I’m a little bit confused about what T1, T2 and T3 means. First, the questionnaire to which the authors are referring should be specified, and then also to this distinction between timepoints (see also the related point 4).

We explained T1, T2 and T3 that it is related only to the questionnaire of life perceptions. See lines 323-326 and legend of Figure 1

8)    P. 6 line 256, p.7 lines 257-258. “Neurotypical adolescents (control group?) reported, on average, higher scores compared to the control group (?)”. There is something wrong in this sentence. Neurotypical adolescents = control group vs. adolescent with Asperger’s syndrome = clinical group. So, please check this sentence.

 There is a mistake. We corrected it. See lines 355-357.

9)    P. 8, lines 267-269 “However, in terms of the adaptive behavior of Asperger's children assessed by their parents, the worst adaptive behaviors were several, especially the area of socialization, 268 evaluated by all parents as the area with the worst adaptive behavior (Figure 4)”. English check is needed.

We rewrote it to clarify. See lines 378-380.

10) In general, graphics must be improved.

 We improved them, explaining the chosen colors and with legends.

Discussion

11) P.8, line 296 “taking into consideration the entire family unit” I’m not sure that this is true, because usually who complete the questionnaire (online) is the mother (and children for this specific study), unless the instructions given to the participants were that both parents should complete the questionnaires (see point 4).

We changed it. See line 407

Reviewer 2 Report

I consider it extremely important to demonstrate the effects on the lives of smaller populations such as people with Asperger's syndrome during the pandemic infection caused by Covid-19. The authors adequately highlight the fact that, in the case of people already struggling with anxiety, the quarantine procedures and the later opening of schools may have resulted in a serious psychological burden for this group. However, as the authors also note, the extremely low number of people and the age group (teenagers) who already have serious mental problems are, in my opinion, not sufficiently representative of people with Asperger's syndrome. The really big advantage of the publication is that the tests of the people included in the study come from the region most affected by the pandemic, however, a more comprehensive picture cannot be formed, unfortunately, due to the extremely limited number of staff. The results tend rather than show clear differences.

I consider it a further problem that it only focuses on the child-father relationship, and again due to the small number of items, the coping strategies of the families cannot be assessed in action, since the heterogeneity of the samples would show very significant differences in the case of such a small amount. Thus, he only tangentially mentions solutions that some families used to overcome the disease, and was unable to include them in his investigations.

In addition, I find it very problematic that the interpretability of the figures does not reach the level that suits me. It is difficult to understand, the captions on the figures do not or little help understanding. I recommend their modification.

Author Response

I consider it extremely important to demonstrate the effects on the lives of smaller populations such as people with Asperger's syndrome during the pandemic infection caused by Covid-19. The authors adequately highlight the fact that, in the case of people already struggling with anxiety, the quarantine procedures and the later opening of schools may have resulted in a serious psychological burden for this group. However, as the authors also note, the extremely low number of people and the age group (teenagers) who already have serious mental problems are, in my opinion, not sufficiently representative of people with Asperger's syndrome. The really big advantage of the publication is that the tests of the people included in the study come from the region most affected by the pandemic, however, a more comprehensive picture cannot be formed, unfortunately, due to the extremely limited number of staff. The results tend rather than show clear differences.

We added these right considerations in the discussion. See lines 539-541.

I consider it a further problem that it only focuses on the child-father relationship, and again due to the small number of items, the coping strategies of the families cannot be assessed in action, since the heterogeneity of the samples would show very significant differences in the case of such a small amount. Thus, he only tangentially mentions solutions that some families used to overcome the disease, and was unable to include them in his investigations.

We added your important consideration as recommendations for future research in the discussion. See lines 596-598.

In addition, I find it very problematic that the interpretability of the figures does not reach the level that suits me. It is difficult to understand, the captions on the figures do not or little help understanding. I recommend their modification.

We changed them adding also legends.